# Bacterial Biofilm Formation on Nano-Copper Added PLA Suited for 3D Printed Face Masks

**DOI:** 10.3390/microorganisms10020439

**Published:** 2022-02-14

**Authors:** Annika Kiel, Bernhard Peter Kaltschmidt, Ehsan Asghari, Andreas Hütten, Barbara Kaltschmidt, Christian Kaltschmidt

**Affiliations:** 1Department of Cell Biology, Faculty of Biology, Bielefeld University, 33615 Bielefeld, Germany; annika.kiel@uni-bielefeld.de (A.K.); ehsan.asghari@uni-bielefeld.de (E.A.); barbara.kaltschmidt@uni-bielefeld.de (B.K.); 2Department of Thin Films and Physics of Nanostructures, Center of Spinelectronic Materials and Devices, Faculty of Physics, Bielefeld University, 33615 Bielefeld, Germany; b.kaltschmidt@uni-bielefeld.de (B.P.K.); andreas.huetten@uni-bielefeld.de (A.H.)

**Keywords:** personal protective equipment, face masks, 3D printing, biofilm, antimicrobial, PLA, nano-copper, *E. coli*, *S. aureus*, *P. aeruginosa*

## Abstract

The COVID-19 Pandemic leads to an increased worldwide demand for personal protection equipment in the medical field, such as face masks. New approaches to satisfy this demand have been developed, and one example is the use of 3D printing face masks. The reusable 3D printed mask may also have a positive effect on the environment due to decreased littering. However, the microbial load on the 3D printed objects is often disregarded. Here we analyze the biofilm formation of *Pseudomonas aeruginosa*, *Staphylococcus aureus,* and *Escherichia coli* on suspected antimicrobial Plactive™ PLA 3D printing filaments and non-antimicrobial Giantarm™ PLA. To characterize the biofilm-forming potential scanning electron microscopy (SEM), Confocal scanning electron microscopy (CLSM) and colony-forming unit assays (CFU) were performed. Attached cells could be observed on all tested 3D printing materials. Gram-negative strains *P. aeruginosa* and *E. coli* reveal a strong uniform growth independent of the tested 3D filament (for *P. aeruginosa* even with stressed induced growth reaction by Plactive™). Only Gram-positive *S. aureus* shows strong growth reduction on Plactive™. These results suggest that the postulated antimicrobial Plactive™ PLA does not affect Gram-negative bacteria species. These results indicate that reusable masks, while better for our environment, may pose another health risk.

## 1. Introduction

In the COVID-19 pandemic, the demand for medical devices such as face masks to reduce viral spread has increased enormously. The use of face masks is recommended by the WHO when meeting with other people [1]. A recent review has provided evidence that the wearing of masks reduced the transmission of infected respiratory particles by filtering [2]. While the priority of a face mask is to avoid viral delivery, the potential risk of microbial contamination for the mask wearer should not be underestimated. 

There is a strong demand for medical devices, resulting in shortages in personal protective equipment in many countries. There has been tremendous pressure on hospitals and researchers to find new and reusable protective equipment. The need for equipment in the medical field and the effect of single-use supplies on the environment demands different approaches for personal protective equipment.

The application of single-use face masks conflicts with current increased environmental concern. The amount of municipal waste generated by discarded single-use face masks and other personal protective equipment has increased enormously. Three billion masks are discarded globally each day, which has created major environmental problems [3]. People tend to dispose of their face masks improperly, which leads to increased littering [4]. This results in many other environmental problems; for example, a single mask can release 1.5 million microplastic particles [5,6].

The medical shortages and the environmental concerns call out for new developments of multi-use personal protective equipment. Researchers from various fields joined forces to create new equipment to close the gaps in needed medical supplies. One promising field turned out to be using three-dimensional (3D) printed objects produced by additive manufacturing. 3D printing has already become a useful tool in the medical field, especially for the design and fabrication of prosthetics [7,8,9]. This technique is also a cost-effective and fast method for the rapid prototyping of personal protective equipment like face masks. Open source instructions can be found and make it easy for everyone to print 3D face masks [10]. For this study, the model of the so-called “Montana” mask pioneered by Dustin Richardson, Spencer Zaugg, and Colton Zaugg at the Billings Clinic in Montana [10] was used. In Figure 1, the production cycle for a 3D-printed face masks is depicted. The face mask consists of a 3D-printed face mask and a replaceable filter (Figure 1, right pictogram white piece within the mask). For example, one certified single-use medical face mask can be cut into six pieces which then can be used as a filter for the here shown white inside of the 3D-printed face mask. This model counteracted the littering problem by a factor of six. Anyone with access to a 3D printer can print masks to meet the strong demand [11].

A wide range of 3D printing materials is commonly available. Especially when used for medical devices, sterility and antimicrobial properties play an important role. Therefore the demand for antimicrobial properties has also entered this 3D filament field. Some companies already sell filaments provided with antimicrobial properties [12].

In this study, we have chosen the antimicrobial Plactive™ Filament by Copper 3D. The Plactive™ PLA has a nano-Copper additive, for which antimicrobial action could be scientifically validated [13]. Zuniga and coworkers have also shown the successful use of Plactive™ for 3D-printed prostheses. Therefore Plactive™ material could turn out to be a promising material for the use of 3D-printed face masks. The face, especially the area around the mouth, is a source of many microorganisms such as bacteria like *Pseudomonas aeruginosa* [14,15]. However, only a few studies have investigated the biofilm formation potential on such materials [12,16]. In contrast to these studies, we use biological and physical techniques such as scanning electron microscopy (SEM) to investigate the formation of biofilms. This analysis generates ultrastructural information of the growth phase of bacteria and allows the resolution of up to a single bacterium. To our knowledge, no other group has investigated the ultrastructure of biofilms on 3D printed PLA objects before.

In contrast to their planktonic counterparts, biofilms form a highly structured and organized community with functional heterogeneity. Bacteria in biofilms can form an extracellular polymeric substance (EPS) matrix, shielding them from harmful external threats, such as antibiotics and disinfectants [17,18,19]. Compared to the planktonic life form, biofilms can be up to 1000× more resistant to antibiotics and antimicrobials [20,21,22]. Therefore the development of antibacterial surfaces to prevent biofilms is particularly important in the medical field. 

To study antimicrobial effects of 3D printing filaments on biofilm formation, we selected three bacterial species: *Pseudomonas aeruginosa*, *Staphylococcus aureus,* and *Escherichia coli*, because these species are among the major groups that colonize human skin and hair [14,15]. In contrast to other studies, the *P. aeruginosa* strain was isolated from domestic washing machines [23] and therefore represented a real-life germ. Furthermore, these bacteria cause many diseases, such as hospital acquired pneumonia, nosocomial bloodstream infections, diarrheal infections, meningitis, wound infections, and septicemia [24,25,26,27].

We analyzed the initial biofilm formation stages on 3D printing materials by ultrastructural scanning electron microscopy to present typical biofilm characteristics. Furthermore, we analyzed the vitality of the attached cells by live and dead staining using confocal laser scanning microscopy (CLSM), which gave additional evidence regarding the bacteria viability. In addition, the direct quantification method used to determine the number of viable cells by plate counting (colony-forming unit assay) was performed by detaching the bacteria adhering to the tested materials.

## 2. Materials and Methods

### 2.1. D Printing and Used Filaments

The 3D model was created in FreeCAD (open source software) and then loaded in the PrusaSlicer software and printed using a Prusa i3 MK3S+ (Prusa Research, Prague, Czech Republic) 3D Printer. Two different PLAs were used for fabrication. Plactive™ (Pactive™ 1% Antibacterial Nanoparticles, Copper 3D, Santiago, Chile) with antimicrobial activity and for comparison a white standard PLA (Giantarm™ by Geetech, Shenzhen, China). The samples were printed at 15% infill, at a temperature of 210 °C with a heat bed temperature of 60 °C and 0.15 mm layer height. The print speed was 50 mm/s, and the travel speed was 180 mm/s. The samples used for analysis had dimensions of 1 × 1 × 0.4 cm. Before samples were examined microbiologically, they were swiped with Isopropanol, and each side was UV sterilized for 30 min. A sterility control was performed for all tested samples by subsequently incubating the samples in Luria-Bertani (LB) medium for 24 h and spreading the supernatants onto LB agar plates.

### 2.2. Bacterial Species

Three bacterial species were selected for the biological assays. The strong biofilm producer *Pseudomonas aeruginosa*, isolated from a domestic washing machine [23], *Staphylococcus aureus* (DSMZ 24167), and *Escherichia coli* XL1-blue strain (Stratagene/Agilent, San Diego, CA, USA). The three bacterial strains were inoculated from frozen stock kept under −80 °C and cultured for 24 h on fresh LB agar plates.

### 2.3. Scanning Electron Microscopy (SEM)

For an ultrastructural observation of biofilms grown on Plactive™ and Giantarm™ PLA samples, we used scanning electron microscopy (SEM). The Plactive™ and standard PLA Giantarm™ samples were bedded on two filter papers soaked with 1.5 mL physiological saline (0.9% NaCl) in a 6 well plate. Since the surface sides differ slightly due to the 3D printing process, we used the bottom side for all experiments. The three bacterial species were inoculated from fresh LB agar plates and pre-cultured in 10 mL LB medium overnight. For each bacterium, three printed samples from each material were inoculated with 80 µL bacterial suspension, adjusted at OD600 = 0.001. To spread the droplet along the surface, we used a plastic cover film with the size of 0.8 × 0.8 cm placed on top. The samples were incubated for 24 h at 37 °C in static conditions. After the incubation period, planktonic cells were washed away by submerging the samples two times in physiological saline (0.9% NaCl) followed by one time in bidest H_2_O. The samples were fixed with half-strength Karnovsky’s solution (2% paraformaldehyde, 2.5% glutaraldehyde) for 30 min. The fixed samples were dehydrated using 50, 70, 80, 90, 95, and 100% (*v*/*v*) graded ethanol. To improve the conductivity of the samples, they were sputter-coated with a layer of 4 nm Ruthenium. For examination, a Helios NanoLab DualBeam 600 (FEI Company, Hillsboro, Oregon, United States) scanning electron microscope (SEM) was used. For image acquisition, the microscope parameters were set to an acceleration voltage of 2 kV and a beam current of 0.17 nA. This was necessary to avoid the destruction of the samples by the electron beam. To better visualize the 3D structures of the biofilm, images were both taken from the top view and a side view. Side view images were taken by tilting the samples by 52 degrees.

The size measurement of the bacteria was done by using the open source FIJI (ImageJ) software. For rod shaped bacteria, length and width were measured with the straight line tool. In the case of *S. aureus,* circles were size-matched to the bacteria, and the area of the circles was measured. The diameter was then calculated from the area.

### 2.4. Confocal Laser Scanning Microscopy (CLSM)

To visualize viable bacteria biofilms, LIVE/DEAD staining followed by confocal laser scanning microscopy was performed. Biofilms were grown on 3D printed samples as described in the scanning electron microscopy (SEM) section. After an incubation period of 24h, the samples were washed three times in physiological saline (0.9% NaCl). The biofilm formation was analyzed using the FilmTracer™ LIVE/DEAD Biofilm Viability Kit (Molecular Probes, Invitrogen, Carlsbad, CA, USA) according to the manufacturer’s instructions. The SYTO 9 green fluorescent nucleic acid stain measured at 482 nm excitation and 500 nm emission was used to visualize live bacteria with an intact cell membrane. Bacteria with a compromised membrane, considered dead or dying, were stained with propidium iodide (red), measured at 490 nm excitation and 635 nm emission. To obtain a percentage of live and dead cells, five images of the same magnification were evaluated with FIJI (ImageJ) software for each material using the following routine: First, the images were converted to binary images. Then the amount of white and black pixels for each image was calculated with the FIJI selection tool. The last step was to compare the number of pixels of the living and dead cells to the background pixel values.

### 2.5. Antibacterial Testing with Colony-Forming Unit Assay (CFU)

To test the antibacterial activity of PlactiveTM in terms of the viable colonies, the colony-forming unit assay was performed as a direct quantification method. Biofilms were grown on 3D printed Plactive™ and Giantarm™ PLA samples as described in the scanning electron microscopy (SEM) section. Afterward, planktonic cells were washed away by submerging the samples three times in physiological saline (0.9% NaCl). The biofilm was detached from the surface by vigorous vortexing for at least 1 min (Vortex Genie, Fisher Scientific, USA) in 10 mL physiological saline (0.9% NaCl). A serial dilution series was performed, and 100 µL of each dilution was plated onto fresh LB agar plates. The plates were incubated overnight at 37 °C. After incubation, colonies were counted and the colony-forming units were calculated.

### 2.6. Statistical Analysis

For all statistical analysis an unpaired nonparametric Mann-Whitney Rank test was performed with P value style GP with: 0.1234 (ns), 0.0332 (*), 0.0021 (**), 0.0002 (***) and <0.0001 (****). Error bars always represent the standard error of mean (SEM) in all experiments.

## 3. Results

### 3.1. Ultrastructural Investigation of Biofilm Formation on 3D Printed Samples

We analyzed the biofilm formation of *Pseudomonas aeruginosa, Staphylococcus* aureus, and *Escherichia coli* on Plactive™ and Giantarm™ PLA 3D printed samples by SEM. In addition to the ultrastructural observation of the biofilms, we performed a size measurement of the bacteria to draw initial conclusions about the viability of the bacteria. The gram-negative strain *P. aeruginosa* shows an adhesion distributed over the entire surface and microcolony formation on both 3D printed materials (Figure 2A). These ultrastructural images show that *P. aeruginosa* undergoes the initial stages of biofilm maturation after only 24 h on both materials. These stages are irreversible attachment, formation of a monolayer, EPS matrix formation, and the beginning of microcolony formation. In the case of *P. aeruginosa,* the beginning of the formation of protective slime can also be seen in the bottom part of Figure 2A Giantarm™ PLA. The size measurement of the bacteria reveals that bacteria adhered to Giantarm™ PLA are slightly longer (0.98 µm ± 0.16 µm) in comparison to the bacteria adhered to the Plactive™ (1.26 µm ± 0.18 µm) material. The width measurement, on the other hand, shows overlapping sizes (Plactive™: 0.41 ± 0.05 µm, Giantarm™ PLA: 0.55 µm ± 0.07 µm).

*Staphylococcus aureus* is a Gram positive bacterium that characteristically and eponymously forms coccoid spheres. Compared to the images of *P. aeruginosa* on the Plactive™ 3D material, it can be seen that *S. aureus* is more likely to adhere to the surface in a scattered manner (Figure 2A and Figure 3A Plactive™). For both materials, it can be observed that holes and grooves in the surface, produced due to the printing process, are more likely colonized by *S. aureus*. Inside these holes and grooves, monolayers and horizontal growth can be observed. The diameter measurements of *S. aureus* bacteria show only slight differences in the two tested materials (Figure 3B, Plactive™: 0.70 µm ± 0.08 µm, Giantarm™ PLA: 0.73 µm ± 0.11 µm).

*Escherichia coli* as another Gram negative bacteria strain reveals similar growth compared to *P. aeruginosa*. The bacteria adhere distributed on the entire surface of both materials. Only a few colonies were found that performed horizontal growth. There is no visible effect of the Plactive™ material on the adhesion of *E. coli*. The length measurement reveals that bacteria adhered to the Plactive™ material are slightly longer (Figure 4B, Plactive™: 2.15 µm ± 0.39 µm, Giantarm™ PLA: 1.98 µm ± 0.38 µm). The width measurements show no differences between the tested materials.

To sum up, all three tested bacteria species adhered to both 3D printing materials. *P. aeruginosa* was also already able to undergo the first stages of biofilm formation on both materials. The ultrastructural observation of the adhesion of the tested bacteria could not reveal any significant differences in the adherence potential of the tested materials for the Plactive™ and Giantarm™ PLA 3D filaments. Here no antimicrobial effect of the Plactive™ material could be observed.

### 3.2. Analysis of Biofilm Viability on 3D Printed Samples

#### 3.2.1. LIVE/DEAD Observation of Biofilm

To underline the SEM results, biofilms were again grown on the 3D samples and examined by confocal laser scanning microscopy. The results of the LIVE/DEAD staining reveal that vital biofilms could form on both 3D printing filaments (Figure 5). The percentage evaluation of live and dead cells, calculated by the sum of the total stained cells, shows that attached bacteria represent both dead and living cells (Figure 5B,E,H). The ratio of live and dead cells for *P. aeruginosa* and *S. aureus* was about 1:1 for both tested materials (Figure 5B,E). Also, we analyzed the ratio of the area covered by live and dead cells related to the uncovered surface to better understand the amount of live and dead cells attached to the tested 3D printed materials (Figure 5C,F,I). For *P. aeruginosa*, almost wave-like colonizations are recognizable on the Plactive™ material (Figure 5A Plactive™). Also, on Giantarm™ PLA samples, attached live and dead cells could be visualized (Figure 5A Giantarm™ PLA). The colonization between the Plactive™ samples and the Giantarm™ PLA samples does not show significant differences even when comparing total live and dead cells (Figure 5B Plactive™ and Giantarm™ PLA). The attached *P. aeruginosa* cells, or biofilms in the initial stages, show a higher tendency of dead cells for both tested samples (Figure 5B). However, if the covered area of the tested materials is brought into account, *P. aeruginosa* attaches nearly three times more often to the Plactive™ materials (Figure 5C, 10.6% live cells on Plactive™, 11.5% dead cells on Plactive™, 3.3% live cells on Giantarm™ PLA and 3.7% dead cells on Giantarm™ PLA). The Gram positive bacteria *S. aureus* does not show such strong uniform growth on the tested surfaces compared to the other two species (Figure 5C). Instead, the colonies form rather small bulks/clusters. These results are consistent with the SEM images (Figure 3A). As mentioned before, the ratio of live and dead cells for *S. aureus* is 1:1 in relation to the total number of bacteria, but in contrast to the results of *P. aeruginosa*, with a higher tendency of live cells (Figure 5E). Considering the total covered area of live and dead cells, the colonization on the Giantarm™ PLA sample is almost twice as high in comparison to the Plactive™ samples (Figure 5F, 2.4% live cells on Plactive™, 2.5% dead cells on Plactive™, 4.9% live cells on Giantarm™ PLA and 4.3% dead cells on Giantarm™ PLA). The CLSM results of *E. coli* show strong uniform growth on both tested materials (Figure 5G), which could already be observed in the SEM images (Figure 4A). Considering the total *E. coli* bacteria attached to the surfaces, a higher tendency of dead cells on both materials could be observed (Figure 5H, 44.5% live cells on Plactive™, 55.5% dead cells on Plactive™, 37.1% live cells on Giantarm™ PLA, and 62.9% dead cells on Giantarm™ PLA). For the Plactive™ samples even with highly significant tendencies. The results for the total covered area compared to the uncovered background also show a higher tendency of dead bacteria on both materials (Figure 5I, 10.6% live bacteria on Plactive™, 13.4% dead bacteria on Plactive™, 18.4% live bacteria on Giantarm™ PLA, and 22.9% dead bacteria on Giantarm™ PLA). On Giantarm™ PLA, *E. coli* reveals the highest coverage of all tested bacteria. Taken together, also for these results, the antimicrobial properties of Plactive™ have no apparent influence on the attachment of *P. aeruginosa* and *E. coli* and therefore on possible biofilm formation. Only for *S. aureus* could a 50% reduction be observed when looking at adhered bacteria compared to the surface background. These results might indicate that the Plactive™ material has an antimicrobial effect on the Gram positive bacterium *S. aureus*.

#### 3.2.2. Direct Quantification of Viable Cells within the First Stages of Biofilm

To directly quantify the viable cells attached to the tested material surfaces, we performed the colony-forming unit assay. The number of living colonies for *P. aeruginosa* did not differ significantly between the tested materials (Figure 6A), with a mean of living colonies of 3.9 × 10^7^ on Plactive™ and 3.8 × 10^7^ on Giantarm™, again revealing that the biofilm-forming potential of *P. aeruginosa* is not affected by the antimicrobial properties of Plactive™. Likewise, for *E. coli*, no significant influence of the antimicrobial properties of the Plactive™ material could be detected (3.5 × 10^5^ living colonies on Plactive™ and 4.5 × 10^5^ living colonies on Giantarm™ PLA). Only for the Gram positive bacteria, *S. aureus,* could a highly significant effect on the viability be observed, with a mean of living colonies of 3.1 × 10^6^ on Plactive™ compared with 3.6 × 10^7^ on Giantarm™ PLA. This antimicrobial effect against *S. aureus* could also be observed in the previously described comparison between the covered area and the uncovered background by live and dead staining.

## 4. Discussion

During the Coronavirus pandemic, a shortage of personal protection equipment occurred worldwide (WHO 2020). The introduction of mask requirements in areas of everyday life, such as grocery shopping or the use of local transport, quickly led to shortages of face masks. Novel strategies were developed for sustainable and reusable 3D printed face masks to counteract the bottlenecks that have arisen [10,28,29]. To avoid bacterial growth on 3D printed materials, several metal blended products appeared on the market [12]. Here we have analyzed biofilm growth on polylactide acid-derived substrates for 3D printing such as unblended PLA from Giantarm™ and Copper blended Plactive™. Copper has historically been known to have antimicrobial activity. Even in ancient Egypt, copper was used to preserve water and food, and also for medical purposes [30,31,32,33]. Today, copper is used in many medical approaches; in birth control, for example, as a copper IUD or copper chain [34]. Some studies propose that the use of copper as an antimicrobial agent in hospitals could greatly reduce hospital-acquired infections (HAI). Copper alloying of frequently touched surfaces such as door handles, door push plates, or tap handles in hospitals revealed a significantly lower amount of bacteria compared to the standard items [35]. Even in the fight against the current pandemic, copper seems able to inactivate COVID-19 viruses [36]. Even though the use of copper reveals numerous advantages, metal nanoparticles, as in the Plactive ^TM^ filament tested here, could also pose a potential health risk [37]. Metal nanoparticles can enter the human body through the respiratory tract or the digestive and dermal systems [38]. Neurological changes caused by nanoparticles that penetrate the nervous system via the olfactory bulb could be observed in rats [39,40], indicating that they may also pose a risk to humans.

We detected a significant reduction of Gram positive *S. aureus* biofilms on Plactive™. But by using SEM and CLSM, we were able to show that all Gram negative bacteria show uniform growth on all tested substrates. SEM analysis demonstrated a multilayered biofilm for *P. aeruginosa* and *E. coli* with already typical features of the initial maturation stages during biofilm growth [23]. However, *S. aureus* did not grow in surface covering biofilms but formed typical grapevine-like bulks called *staphylos* in greek [41]. *S. aureus* preferred to attach to inside grooves and cavities caused by the 3D printing process. The size measurements of *S. aureus* and *E. coli* bacteria attached to Plactive™ and Giantarm™ PLA, listed in Table (see Figure 7A), revealed no significant differences between the tested 3D printing filaments (see Figure 7B). Interestingly *P. aeruginosa* showed significantly larger cell sizes on Giantarm™ PLA, but still in the range of literature data. The length of *P. aeruginosa* was around 1.1 µm, and the width was around 0.5 µm, as determined from scanning electron microscopy. The length, width, and diameter of all tested bacteria are in the range of literature data listed in Table (see Figure 7A). Slightly smaller sizes in our measurements could be due to probe preparation by drying for the SEM or might be species-specific for our wild-type strain *P. aeruginosa*. Here *S. aureus* is the smallest of the analysed bacteria with a diameter of around 0.72 µm. *E. coli,* with an averaged length of 2.1 µm, is the largest bacterium measured here. The SEM images in Figure 4 show rods of very different lengths. The relatively long rods are probably shortly before cell divisions. The width of *E. coli* with 0.6 µm fits together with the determined length in the range found in the literature.

In addition, we analyzed the viability of the bacteria by LIVE/DEAD staining and colony-forming unit assays (CFU). Live and dead bacteria appeared nearly at the same amounts (1:1 ratio) on all tested materials. Only for *E. coli* was the tendency for dead bacteria on Plactive™ significantly higher. A similar low viability of *P. aeruginosa* was reported for gentamycin-loaded bone cement (47%), whereas plain bone cement depicted biofilms with 91% viable bacteria [42]. We expected higher viability on plain PLA without copper, but this was observed only for *E. coli*. Interestingly the ratio of the covered area by live and dead cells in relation to the uncovered surface revealed a higher tendency of dead cells on the Plactive™ material for *S. aureus* and *E. coli*. The colony-forming unit assay results for *S. aureus* underlined these results, with high significant reduction rates on the Plactive™ material. A closer look at the CFU results for *E. coli* showed that the number of living colonies was a hundred times lower than *P. aeruginosa*. However, *E. coli* revealed the highest amount of total surface colonization compared to the other tested bacteria. The comparably low amounts of living colonies should be considered in the context of the cell size of the tested bacteria. The size measurements shown in the previously described SEM results of all tested bacteria revealed that *E. coli* bacteria form rods approximately twice the size of *P. aeruginosa*. The living colony count of the gram-negative *P. aeruginosa* reveals that the here used wild-type strain is capable of vital growth on both materials. In contrast to other germs, *P. aeruginosa* covers more surface on Plactive™ in comparison to Giantarm PLA™ (See Figure 5C). In addition, we see an increase in stress-induced growth. Elevated concentrations of heavy metals like copper in the environment caused by human activities create enormous selection pressure on the present microorganisms [43]. Even though copper is an important cofactor for many enzymes, high levels of copper are toxic [44,45,46]. The bacteria commonly found in water and soil, such as *P. aeruginosa*, have several mechanisms for coping with heavy metal stress [47,48,49]. *P. aeruginosa* produces an exopolysaccharide capsule, which is composed primarily of alginate protecting the bacterium [50,51]. Another method used by *P. aeruginosa* to reduce stress caused by heavy metals such as copper is the reduction or oxidization to less toxic forms [52,53]. Some bacteria strains are also capable of activating efflux by transporting metal cations out of the cytosol and periplasmic space [44,47,54]. This active transport is mediated by a network of different transporter families [44,54,55]. One member of this family is the soft metal P-type ATPases CopA, which is well studied in *E. coli* but can also be found in other bacteria. CopA effluxes Cu ^+1^ actively out of the cell [56,57]. The high tolerance to copper nanoparticles of Plactive^TM^ 3D printing material observed here for both gram-negative bacteria tested may be due to the strategies mentioned above. The stressed-induced growth of *P. aeruginosa* can be explained by the fact that in contrast to *E. coli* and *S. aureus*, *P. aeruginosa* is a germ that frequently occurs in the environment and therefore often has to cope with heavy metal contamination. It should also be noted that the *P. aeruginosa* isolate used here is a wild type strain isolated from a household washing machine, and therefore is adapted to harsh environmental conditions.

Reduction in growth rates and cell death of the Gram positive *S. aureus* caused by copper could be already observed in other studies [35,58,59]. Copper ions released by copper alloy surfaces led to cell death in both Gram positive and Gram negative bacteria [60]. However, even the results of Meyer and coworkers show that *P. aeruginosa* survives longer (up to 90 min) on copper alloying surfaces compared to *S. aureus* (up to 60 min). They also show that the loss of membrane integrity in Gram positive and Gram negative strains reveals critical differences [60]. Revealing that the different membrane compositions of Gram positive and Gram negative bacteria could be the origin of these effects and should be further investigated in the context of antimicrobial properties like copper in the future.

3D-printed materials are widely developed for medical applications, such as prostheses for hands, fingers, or even upper limbs [61,62,63,64]. They are much more cost-efficient than conventional prostheses [63,64,65]. However, our analysis shows only a limited inhibition of bacterial biofilms on Plactive™ manufactured by Copper 3D. Inhibition of bacterial growth was limited on Gram positive bacteria such as *S. aureus*. This could be due to the different bacterial cell walls. Gram negative biofilms were positively related to antimicrobial resistance, such as resistance to gentamicin and zephtamycine in *E. coli,* and to Cyprofloxacin in *P. aeruginosa* [66]. The authors conclude that the acquisition of specific antimicrobial resistance can enhance biofilm formation in several species of Gram negative bacteria. Thus there is a strong need for 3D printing substrates with antimicrobial activity to counteract antibiotic resistant bacteria. The study of Hall Jr. and coworkers compared the biofilm growth on 3D-printed materials with different proposed antimicrobial additives. They revealed that polymers containing 40% metal show the greatest antimicrobial and antibiofilm activities. In contrast, the results presented here clearly reveal differences in antimicrobial activity against Gram negative and Gram positive bacteria. Also, this study and our previously published study [23] show that wild-type bacterial strains are far more resistant to external influences and show higher biofilm forming potential than their strain collection counterparts.

## 5. Conclusions

Environmental protection and sustainability are becoming increasingly important aspects of our everyday lives. Innovative ideas to bring sustainability into our everyday lives are currently being developed worldwide. The abolition of single-use products like plastic straws and plastic cutlery in Germany [67] is a step in the right direction. Here we analyzed a 3D printing material as a substitute for single-use face masks: Plactive™. Compared to unblended PLA (Giantarm™), Plactive™ material is only antimicrobial for Gram positive *S. aureus* but not for Gram negative *P. aeruginosa* and *E. coli*. Future studies should concentrate on alternative nanomaterials with broader antimicrobial activity. 

The face mask model presented here is a combination 3D printed mask with a replaceable filter. It should be noted that even if the replaceable filter is combined with a certified mask, the combination with the 3D model presented here has not been certified yet.

## Figures and Tables

**Figure 1 microorganisms-10-00439-f001:**
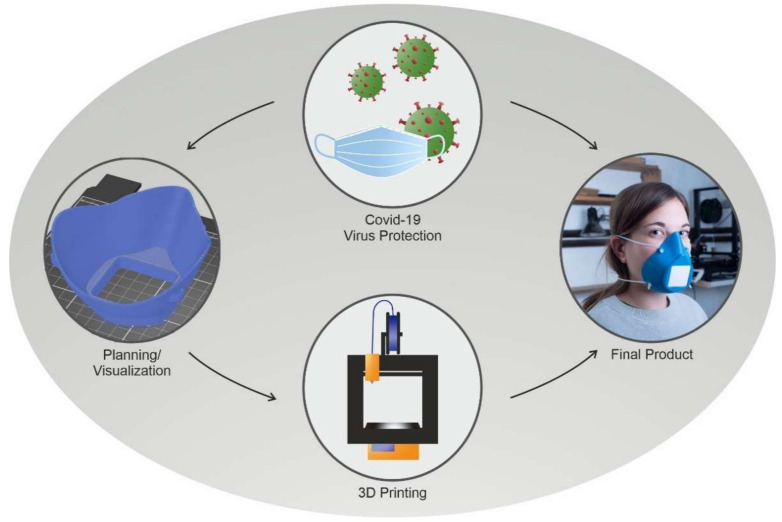
Overview of the development of a 3D-printed face mask. The top image shows pictograms of a typical op mask and three coronaviruses. On the left side the slicer software preview of a 3D printable mask is shown. In the bottom graphic the pictogram of a 3D Printer can be seen. The image on the right shows the finished 3D-printed mask mounted on the face.

**Figure 2 microorganisms-10-00439-f002:**
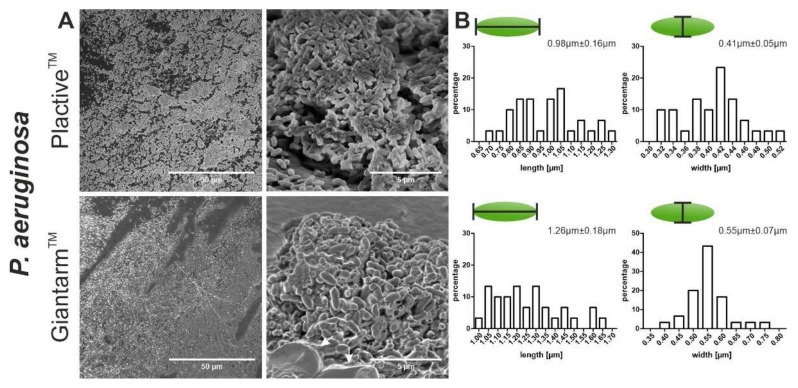
Scanning electron microscopy (SEM) images showing the biofilm formation and size measurement of *P. aeruginosa*. (**A**) The two top SEM images depict the formation of biofilm on Plactive™ PLA. The two bottom images depict the biofilm formation on Giantarm™ PLA. Slime formation is indicated by an arrow. (**B**) The two top graphs visualize the size distribution of the bacteria on Plactive™ PLA, and the two bottom graphs visualize the size distribution of the bacteria on Giantarm™ PLA. For all four graphs, the *y* axis represents the relative percentage and the *x* axis represents the length or width of the bacteria in microns.

**Figure 3 microorganisms-10-00439-f003:**
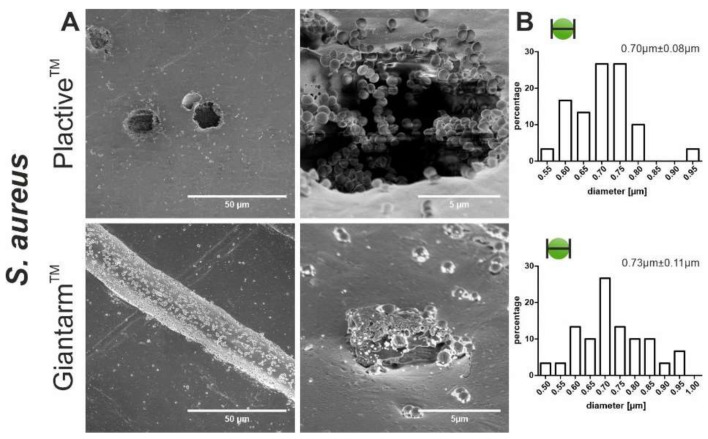
Scanning electron microscopy (SEM) images showing the biofilm formation and size measurement of *S. aureus*. (**A**) The two top SEM images depict the formation of biofilm on Plactive™ PLA, and the two bottom images depict the biofilm formation on Giantarm™ PLA. (**B**) The top graph visualizes the size distribution of the bacteria on Plactive™ PLA, and the bottom graph visualizes the size distribution of the bacteria on Giantarm™ PLA. For all graphs, the *y* axis represents the relative percentage and the x axis represents the radius of the bacteria in microns.

**Figure 4 microorganisms-10-00439-f004:**
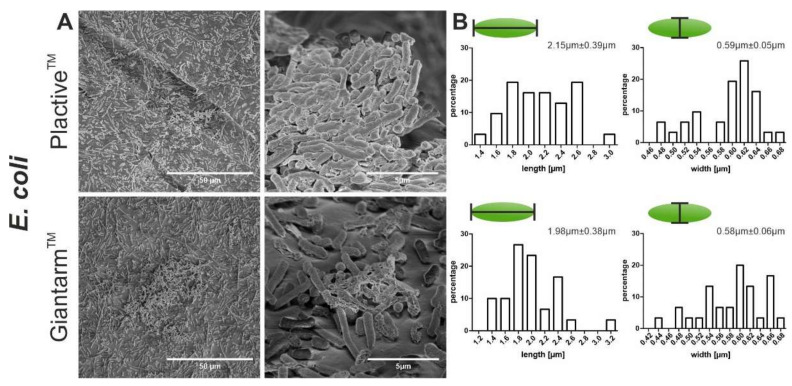
Scanning electron microscopy (SEM) images showing the biofilm formation and size measurement of *E. coli*. (**A**) The two top SEM images depict the formation of biofilm on Plactive™ PLA, and the two bottom images depict the biofilm formation on Giantarm™ PLA. (**B**) The two top graphs visualize the size distribution of the bacteria on Plactive™ PLA, and the two bottom graphs visualize the size distribution of the bacteria on Giantarm™ PLA. For all four graphs, the *y* axis represents the relative percentage and the *x* axis represents the length or width of the bacteria in microns.

**Figure 5 microorganisms-10-00439-f005:**
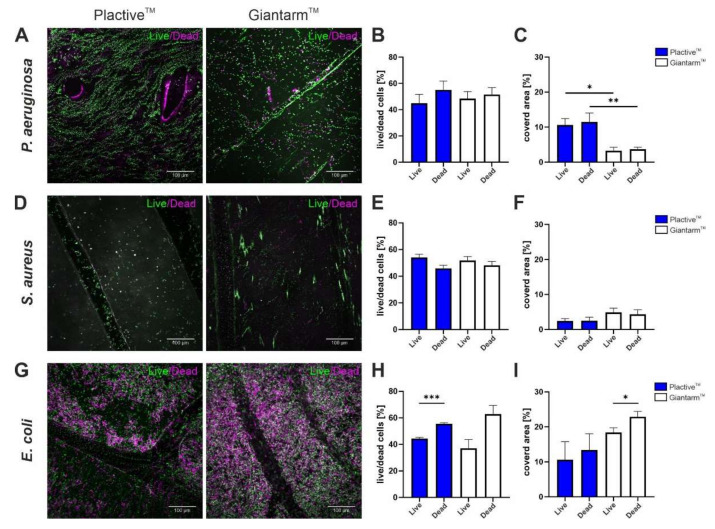
Confocal laser scanning microscope (CLSM) images, percentages of live versus dead cells and percentages of covered area for *P. aeruginosa*, *S. aureus,* and *E. coli*. (**A**) CLSM images of *P. aeruginosa* biofilm formation on Plactive™ PLA (left) and Giantarm™ PLA (right). (**B**) Percentages of live and dead cells of *P. aeruginosa* on Plactive™ PLA (blue) and Giantarm™ PLA (white). (**C**) Percentages of the bacterial covered area in relation to the uncovered area of *P. aeruginosa* on Plactive™ PLA (blue) and Giantarm™ PLA (white). (**D**) CLSM images of *S. aureus* biofilm formation on Plactive™ PLA (left) and Giantarm PLA (right). (**E**) Percentages of live and dead cells of *S. aureus* on Plactive™ PLA (blue) and Giantarm™ PLA (white). (**F**) Percentages of the bacterial covered area in relation to the uncovered area of *S. aureus* on Plactive™ PLA (blue) and Giantarm™ PLA (white). Note the strong antimicrobial effect on *S. aureus* of Plactive™ and Giantarm™. (**G**) CLSM images of *E. coli* biofilm formation on Plactive™ PLA (left) and Giantarm™ PLA (right). (**H**) Percentages of live and dead cells of *E. coli* on Plactive™ PLA (blue) and Giantarm™ PLA (white). (**I**) Percentages of the bacterial covered area in relation to the uncovered area of *E. coli* on Plactive™ PLA (blue) and Giantarm™ PLA (white).

**Figure 6 microorganisms-10-00439-f006:**
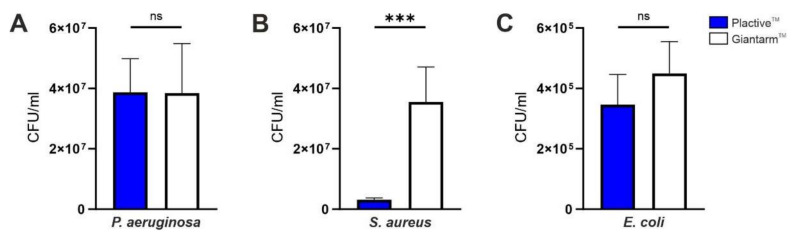
Colony forming unit (CFU) assays of *P. aeruginosa*, *S. aureus,* and *E. coli*. (**A**) CFU assay revealing the number of CFUs per ml (*y*-axis) for the growth of *P. aeruginosa* on Plactive™ PLA (blue) and Giantarm™ PLA (white). (**B**) CFU assay revealing the number of CFUs per ml (*y*-axis) for the growth of *S. aureus* on Plactive™ PLA (blue) and Giantarm™ PLA (white). (**C**) CFU assay revealing the number of CFUs per ml (*y*-axis) for the growth of *E. coli* on Plactive™ PLA (blue) and Giantarm™ PLA (white).

**Figure 7 microorganisms-10-00439-f007:**
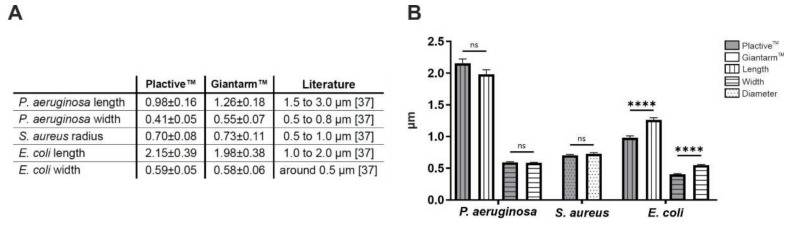
Comparison of the measured and literature size values and mean of the measured size values depicted in a bar graph (**A**) Table showing the measured values and literature values of the three analyzed bacteria (*P. aeruginosa*, *S. aureus*, *E. coli*) [37]. The second and third columns depict the measured values on Plactive™ PLA and Giantarm™ PLA, respectively. The fourth column shows the literature value. (**B**) Bar graph depicting the mean of the measured size values in microns for the three different bacteria. The first group belongs to *P. aeruginosa*, of which the two longer bars show the length of the bacteria on Plactive™ PLA and Giantarm™ PLA, respectively. The two shorter bars visualize the width of the bacteria, the first of the two on Plactive™ PLA and the second on Giantarm™ PLA. The group of two bars in the middle of the graph belongs to *S. aureus*, where the mean of the diameter is visualized, the left bar showing the value on Plactive™ PLA and the right bar the value on Giantarm™ PLA. The four rightmost bars belong to *E. coli*, where the two left bars of the group depict the mean of the length of the bacteria on Plactive™ PLA and Giantarm™ PLA, respectively. The two right bars show the mean of the width of the bacteria on Plactive™ PLA and Giantarm™ PLA.

## Data Availability

All presented data generated by our experiments are included in the manuscript.

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
