# Peer review of "Bacterial Biofilm Formation on Nano-Copper Added PLA Suited for 3D Printed Face Masks"

_microorganisms, 2022, doi:10.3390/microorganisms10020439_

Round 1

Reviewer 1 Report

The publication raises an important topic related to the prevention of infectious diseases. Especially today it is a topical issue as the world is trying to combat the COVID-19 epidemic. The problem of infectious diseases and the use of face masks are interrelated and development of new solutions is needed. However, it should be remembered that COVID-19 is a viral disease and quoting it in a publication related to bacterial infections is not entirely correct. It would have to be verified.

Moreover, there are some editing errors:

  • strains names should be italicized, i.e. lines 19, 74, 89
  • line 34: should be COVID-19 (revise in whole paper)
  • line 74: should be: People tend to dispose their face masks improperly
  • line 93: Sepsis is not a infection but rather a specific condition which arises in response to the infection. Moreover, not only hospital aquired pneumonia can be caused by these bacteria but several other nosocomial infections. Please, revise.
  • line 109, 110: the degree Celsius symbol should be inserted
  • line 112: should be 1.0 x 1.0 x 0.4 cm
  • line 113: Was the sterility control conducted for the filaments? Do Authors prepared the experiments for non-bacteria samples? This should be explained and emphasized.
  • line 132: should be H2O
  • line 134: please unify in whole manuscript "min.", "mins" or "minutes"
  • lines 172, 212, 213: the lack of space before ul
  • lines 303, 306, 307, 309: should be 3.9 x 107 etc.
  • line 376: Authors should discussed this issue

Reviewer 2 Report

Impressive investigation linked with great effort. Results are interesting and manuscript is high quality

However I have some remarks

  • masks must also comply with the legal requirements - therefore, 3D-printing performed by anybody is to be questioned. There is no
  • how is the functionality tested or verified with regard to personnel protection?
  • antimicrobial compounds are of interest but the possible health hazards of such materials should be discussed. Furthtermore, data revealed a low potential against relevant microorganisms-even copper. They might only present  an ad on?
  • for me it is not clear what is ment by ...Out of one single-use face mask six filters can be produced? (page 2)
  • for me it is not clear what the authors mean with face mask-it should be clarified (eg. authors show a typical op mask in the pictogram but the final product does not display such a mask
